# Opportunities and Challenges for the Development of MRCK Kinases Inhibitors as Potential Cancer Chemotherapeutics

**DOI:** 10.3390/cells12040534

**Published:** 2023-02-07

**Authors:** Vanessa M. Ruscetta, Taj J. Seaton, Aleen Shakeel, Stanley N. S. Vasconcelos, Russell D. Viirre, Marc J. Adler, Michael F. Olson

**Affiliations:** Department of Chemistry and Biology, Toronto Metropolitan University, Toronto, ON M5B 2K3, Canada

**Keywords:** cytoskeleton, signal transduction, phosphorylation

## Abstract

Cytoskeleton organization and dynamics are rapidly regulated by post-translational modifications of key target proteins. Acting downstream of the Cdc42 GTPase, the myotonic dystrophy-related Cdc42-binding kinases MRCKα, MRCKβ, and MRCKγ have recently emerged as important players in cytoskeleton regulation through the phosphorylation of proteins such as the regulatory myosin light chain proteins. Compared with the closely related Rho-associated coiled-coil kinases 1 and 2 (ROCK1 and ROCK2), the contributions of the MRCK kinases are less well characterized, one reason for this being that the discovery of potent and selective MRCK pharmacological inhibitors occurred many years after the discovery of ROCK inhibitors. The disclosure of inhibitors, such as BDP5290 and BDP9066, that have marked selectivity for MRCK over ROCK, as well as the dual ROCK + MRCK inhibitor DJ4, has expanded the repertoire of chemical biology tools to study MRCK function in normal and pathological conditions. Recent research has used these novel inhibitors to establish the role of MRCK signalling in epithelial polarization, phagocytosis, cytoskeleton organization, cell motility, and cancer cell invasiveness. Furthermore, pharmacological MRCK inhibition has been shown to elicit therapeutically beneficial effects in cell-based and in vivo studies of glioma, skin, and ovarian cancers.

## 1. Cancer and the Cytoskeleton

Cancer cells undergo numerous modifications that contribute to tumour initiation, growth, and progression [1]. The canonical hallmarks of cancer defined by Hanahan and Weinberg include the activation of invasion and metastasis [2], which depend on multiple changes to the regulation and organization of the actin–myosin cytoskeleton that enable considerable alterations in cell adhesion and motility [3]. In addition, the actin–myosin cytoskeleton makes important, albeit less direct, contributions to additional cancer hallmarks, such as sustaining proliferative signalling and resisting cell death [4]. The mechanisms contributing to cancer-associated changes in actin–myosin organization include cell-intrinsic factors, such as genetic or epigenetic alterations that affect the expression and/or activity of cytoskeleton regulators or structural components, as well as cell-extrinsic influences, including receptor ligands and the physical tumour microenvironment, which stimulate signal transduction pathways that culminate in changes in cytoskeleton organization [5].

## 2. Actin–Myosin Cytoskeleton Regulation

The actin–myosin cytoskeleton in cancer cells is primarily composed of filamentous actin (F-actin) in association with myosin II filaments, which are made up of two heavy and four light chains (two “essential” light chains and two regulatory light chains) [6]. In addition, cross-linking proteins strengthen actin–myosin filaments and help to form specific cytoskeleton structures. The actin–myosin protein complex uses energy derived from ATP hydrolysis to generate actin–myosin contractile force, which promotes the changes in cytoskeleton organization that drive morphological changes and extracellular matrix remodelling to enable altered adhesion and promote cell motility. The phosphorylation of the regulatory myosin light chains (MLC) results in myosin heavy-chain head groups using ATP to move towards the F-actin barbed end in a process called the power stroke [7]. If myosin filaments are associated with >1 actin filament, the power stroke moves the relative positions of the filaments to generate contractile force.

The Rho family of low-molecular-weight GTPases includes key signal transducers that are central to many pathways leading to dynamic cytoskeleton organization [8]. The twenty members of this protein family switch between inactive GDP-bound and active GTP-bound states, aided by GTPase-accelerating proteins (GAPs) that facilitate GTP hydrolysis and guanine–nucleotide exchange factors (GEFs) that promote the exchange of GDP for GTP [9]. The most well-characterized Rho GTPases are RhoA, Rac1, and Cdc42, each of which promotes the formation of distinct F-actin structures. In the active GTP-bound conformation, the GTPases induce varied cellular responses by binding to and regulating the activity of numerous effector proteins. These effectors can be classified into two broad categories with respect to their effects on the actin–myosin cytoskeleton: those that promote actin polymerization and those that catalyse the phosphorylation of MLC.

There are two major types of GTPase-regulated MLC kinases: the Rho-associated coiled-coil kinases ROCK1 and ROCK2 [10] and the myotonic dystrophy-related Cdc42-binding kinases MRCKα, MRCKβ, and MRCKγ [11,12] (Figure 1A). More closely related to the MRCK kinases than to the ROCK kinases is the dystrophia myotonica protein kinase (DPMK). For all six proteins, the kinase domains are similarly positioned in the protein N-termini and are highly related, with ~45–50% overall amino acid identity (Figure 1B). The mapping of the relative amino acid identity at each amino acid for all six kinases onto the three-dimensional structure of MRCKβ illustrates the extensive conservation of the residues proximal to the nucleotide-binding pocket (Figure 2). However, the remainder of the ROCK and MRCK proteins are completely different, including distinct Rho-binding domains in the ROCK proteins and Cdc42/Rac interacting binding (CRIB) domains in MRCK proteins, which enable associations with their cognate Rho family GTPase. Unlike ROCK and MRCK kinases, DMPK is a relatively small protein consisting of only the kinase domain and a DMPK-specific coiled-coil region with no domain that enables interaction with Rho GTPases, making it an outlier in not being a Rho GTPase effector protein. There has been considerably more research on the ROCK kinases over the years, facilitated by the early discovery of selective inhibitors in 1997 [13], followed by the subsequent discovery of many more potent ROCK inhibitors. Consequently, there are many reviews on ROCK that have been published, including the potential utility of ROCK inhibitors for the treatment of cancer [14,15]. In the following sections, the roles of MRCK in normal cells and its associations with cancer will be summarized.

## 3. MRCK Function

The MRCK kinases phosphorylate several proteins that contribute to the regulation of the actin–myosin cytoskeleton. Through the binding of active GTP-loaded Cdc42 to the CRIB domain, the consequent activation of MRCK signalling results in increased MLC phosphorylation [23,24], which may occur through direct substrate phosphorylation or in combination with the phosphorylation of the MYPT1 myosin-binding subunit of the MLC phosphatase complex [25], which results in the inhibition of MLC dephosphorylation. The association of MRCK with the leucine repeat adaptor protein 35a (LRAP35a) promotes the phosphorylation of the myosin II-related protein MYO18A to facilitate MLC-dependent actin–myosin assembly in lamellipodial protrusions in migrating cells, independent of ROCK signalling [26]. Similarly, binding to the leucine repeat adaptor protein 25 (LRAP25) promotes the phosphorylation and activation of LIM kinase 1 (LIMK1) by MRCK, leading to increased cofilin phosphorylation and the consequent inhibition of its F-actin-severing activities [27]. MRCKα had also been reported to Figure 2. Percentage amino acid identities of kinase domains mapped onto MRCKβ. ROCK1, ROCK2, MRCKα, MRCKβ, MRCKγ, and DMPK kinase domains were aligned, and the calculated amino acid identity scores between all six kinases were used to create a heat map on the 3D structure of MRCKβ bound to ADP (black; PDB 4UAK) [20] with UCSF Chimera [21]. The minimum relative amino acid identity threshold was set to 0.5, where any residues below the threshold were coloured white and residues with higher percentage identity scores were coloured in varying shades of red, as indicated phosphorylate and activate LIMK2, although the role of LRAP25 was not examined [22]. Together, these phosphorylation events catalysed by MRCK kinases lead to the stabilization of actin filaments and increased MLC phosphorylation, and the consequent generation of contractile force, which collectively promote cell motility (Figure 3).

It has been reported that, in vitro, the isolated kinase domain of MRCKα autophosphorylates on activation loop Ser234 and Thr240 residues, as well as Thr403 [28]; however, the X-ray crystal structure of the MRCKβ kinase domain [20], as well as the DMPK [29], ROCK1 [30], and ROCK2 [31] kinase domains, revealed that all were in active conformations in the absence of activation loop phosphorylation, suggesting that this post-translational modification is not necessary for the activation of these related kinases. The isolation of full-length proteins from cells lead to the identification of MRCKα Ser1003 [32] and MRCKβ Thr1108 [33] as autophosphorylation sites, but neither appeared to be necessary for kinase activation. As a result, other than the possibility that MRCK autophosphorylation could be used as a biomarker of kinase activity [32], the contribution of these phosphorylations to changes in cell motility has not been formally demonstrated.

## 4. MRCK Expression and Cancer Association

The three members of the MRCK protein family are encoded by separate genes: the gene for MRCKα is Cdc42BPA (chromosomal location 1q42.13), that for MRCKβ is Cdc42BPB (chromosomal location 14q32.32), and that for MRCKγ is Cdc42BPG (chromosomal location 11q13.1). RNA sequencing data collected by The Genotype–Tissue Expression (GTEx) project across 38 tissues from nearly 1000 healthy individuals revealed that MRCKα and MRCKβ are similarly widely distributed, with MRCKβ being more highly expressed (Figure 4A). In contrast, MRCKγ’s tissue distribution is more restricted, with very low levels in approximately one-third of all tissues.

Data from The Cancer Genome Atlas (TCGA) [34,35] pan-cancer atlas compiled from 32 studies indicated that 9% of the 10,953 patients had genomic alterations to one or more of the Cdc42BPA, Cdc42BPB, and Cdc42BPG genes (Figure 4B), with gene amplifications and missense mutations being the most common. Interestingly, microarray analysis identified the increased expression of Cdc42BPA mRNA (identified in the study as PK428) as being significantly associated with an increased risk of breast cancer metastasis [36]. Further examination of the association of Cdc42BPA expression with metastatic breast cancer resulted in it being included in the 70-gene set marketed as Mammaprint [37], which has been thoroughly validated for its clinical utility in prognostic risk assessment [38,39] and treatment decisions [40,41]. The contribution of elevated MRCKα to breast cancer is postulated to be due to its role in promoting enhanced cell motility and F-actin organization [37].

The immunohistochemical analysis identified increased MRCKα and MRCKβ expression in cutaneous squamous cell carcinoma (SCC) relative to normal human skin [32]. Furthermore, microarray analysis of gene expression in glioma patients revealed increased Cdc42BPA mRNA expression relative to normal brain tissue, while immunohistochemical analysis with an antibody that recognized the active autophosphorylated Ser1003 of MRCKα determined that there was elevated kinase activity on tumour margins relative to tumour cores, consistent with increased invasion of glioma cells away from the tumour’s edges [42].

To identify promising kinase targets in high-grade serous ovarian cancer (HGSOC), a multiplexed kinase inhibitor beads and mass spectrometry (MIB/MS) approach was used [43] to profile 25 primary and 10 patient-derived xenograft (PDX) tumour samples [43]. MIBs are composed of multiple pan-kinase inhibitors immobilized on beads that capture protein kinases from cell lysates, which MS then identifies to render a detailed picture of the abundance and activation state of the expressed kinome. In total, 324 kinases were identified, with at least 206 kinases quantified in 70% of the tumours. Both MRCKα and MRCKβ were expressed at significantly elevated levels in HGSOC tumours, and immunohistochemical analysis also revealed moderate/high expression in 87/105 (83%) tumour sections [44]. Consistent with these observations, TCGA data revealed gene amplifications of Cdc42BPA, Cdc42BPB, and Cdc42BPG or their upstream regulator Cdc42 in 96/584 (16.4%) of ovarian cancer cases [45,46] (Figure 5A,B). Furthermore, TCGA microarray analysis of Cdc42BPA mRNA expression in 558 serous ovarian cancer cases indicated that 106 (19%) had log-transformed Z scores greater than 1.0, indicative of elevated transcript levels (Figure 5C). These studies are consistent with the elevated expression and/or activity of the MRCK kinases being a significant contributor to some forms of cancer, with a likely role in the promotion of invasion and metastasis.

## 5. Small-Molecule MRCK Pharmacological Inhibitors

Given the evidence indicating that MRCK proteins likely play roles in cancer growth and progression, there have been several efforts to discover MRCK-selective inhibitors. However, selective MRCK inhibition has been a challenge due to the high homology between the MRCK and ROCK kinase domains (Figure 1 and Figure 2), which can cause potential inhibitors to lack specificity towards the desired target and limit their applications. Nevertheless, several MRCK inhibitors have been described in the literature and can inform the design of future inhibitors.

Chelerythrine chloride was first identified as an MRCKα and MRCKβ inhibitor through the screening of a commercially available panel of 159 protein kinase inhibitors at 10 µM. It was found to reduce the in vitro phosphorylation of recombinant MLC with an IC_50_ of 1.77 µM for MRCKα (Figure 6A) [47,48]. Although chelerythrine chloride was previously reported to be a PKCα inhibitor with an IC_50_ of 0.66 µM [49], no significant inhibition of seven kinases (including highly homologous kinases, such as DMPK, ROCK2, citron kinase, MLC kinase, or even PKCα) was detected. MRCK inhibition was not affected by varying the ATP concentrations, suggesting that MRCK inhibition by chelerythrine chloride proceeds through a non-ATP competitive mechanism, although the mechanism of action has not yet been determined. The treatment of cells with chelerythrine chloride led to changes in F-actin organization and inhibited cell migration, in agreement with an effect on inhibiting MRCK activity [47,50].

In a similar manner, the 159-kinase inhibitor set was screened at 30 µM and 3 µM against MRCKβ using an in vitro assay of peptide phosphorylation [20]. This approach revealed that Y-27632, TPCA-1, and Fasudil exhibited significant inhibitory activity, which was confirmed by a 10-point dose–response analysis for both MRCKα and MRCKβ to determine the IC_50_ values (Figure 6B). The authors suggested that the unexpected inhibition of MRCK by Fasudil might have resulted from the relatively low K_m_ concentration of ATP (0.7 µM) that was used for the in vitro kinase assay. This conclusion is consistent with the previously reported selectivity of Fasudil for ROCK over MRCK that was previously reported when a higher ATP concentration (100 µM) was used for the in vitro kinase assays [51]. The co-crystallization of the MRCKβ kinase domain with Fasudil and TCPA-1 revealed similar positioning, which suggested a mechanism for efficient MRCK inhibition by small molecules.

Due to the high sequence similarity of MRCK and ROCK, it was possible to develop dual ROCK and MRCK kinase inhibitors simultaneously targeting both pathways [52]. A dose-dependent assay conducted with the dual-specificity DJ4 isothiocyanate inhibitor prevented the phosphorylation of MYPT1 on Thr696 by MRCKα and MRCKβ, with IC_50_ of approximately 10 and 100 nM, respectively, and more effectively inhibited ROCK1 (5 nM) and ROCK2 (50 nM) (Figure 6C). The inhibitory effect of DJ4 on ROCK1 and MRCKβ activity was reduced by increasing the concentrations of ATP, consistent with DJ4 acting as an ATP competitor inhibitor. 

The first potent and selective MRCK inhibitor was the 2-pyridyl pyrazole amide inhibitor BDP5290 [53] (Figure 7). BDP5290 was developed from a high-throughput screening campaign that began with 87,225 compounds, followed by iterative rounds of structure–activity relationship (SAR) chemistry, in vitro assays, and X-ray crystallography. It could reduce both breast cancer and squamous cell carcinoma invasion and displayed relatively good potency with affinity concentrations (K_i_) of 10 nM and 4 nM for MRCKα and MRCKβ, respectively. The in vitro selectivity of BDP5290 for MRCKβ was 86 and 46-fold over ROCK1 and ROCK2, respectively, suggesting that there was room for improvement with respect to selectivity. For comparison purposes, at 1 μM ATP, the ROCK inhibitor Y27632 had in vitro IC_50_ values that were merely 16-fold more selective for ROCK kinases over MRCKβ, demonstrating the noteworthy jump in selectivity that BDP5290 achieved. 

The second-generation MRCK azaindole inhibitors BDP8900 and BDP9066 were reported in 2018. They were discovered by starting with fragment-based screening, followed by in vitro assays and X-ray crystallography to enable the SAR chemistry, and had markedly better potency and selectivity, boasting up to 562-fold affinity for MRCK over ROCK, with K_i_ values in the sub-nanomolar range (Figure 7) [32]. BDP8900’s K_i_ values ranged from 0.030 nM to 0.024 nM for MRCKα and MRCKβ, respectively. BDP9066 is the most potent inhibitor to date, with K_i_ values ranging from 0.0136 nM to 0.0233 nM for MRCKα and MRCKβ, respectively. Extensive selectivity profiling against 115 kinases in vitro demonstrated the high selectivity of both BDP8900 and BDP9066, with detailed dose–response analysis of kinase inhibition at their respective ATP K_m_ concentrations indicating that BDP8900 was 43 times more selective and BDP9066 was 27 times more selective for MRCK kinases than any other of the 115 tested. Furthermore, BDP9066 specificity in HGSOC cells was validated by profiling the kinases bound to MIBs in the presence of BDP9066 [44]. By comparing the relative levels of 216 MIB-bound kinases in OVSAHO cells treated with DMSO vehicle or 2 µM BDP9066, the binding of both MRCKα and MRCKβ was found to be reduced >10x more effectively than that of all other kinases, demonstrating the inhibitor’s notable selectivity. Taken together, these studies reveal that it is possible to develop small-molecule MRCK inhibitors with high potency and selectivity.

## 6. Binding Mode of MRCK Inhibitors

Essential interactions between an inhibitor and a kinase can be identified by investigating the binding mode via X-ray crystallography. The most optimized MRCK inhibitor to date, BDP9066, has three distinct parts: an azaindole backbone, a heterocyclic ligand, and a diazaspirocycle (Figure 7). Structural studies revealed that the 7-azaindole backbone is the hinge binder and forms two hydrogen bonds with the carbonyl of Asp154 and amine of Tyr156 (Figure 8A) [32]. Similarly, the pyridine pyrazole group in BDP5290 acts as the hinge binder, forming hydrogen bonds with the backbone of Asp154 and Tyr156 (Figure 8B) [52]. The co-crystal structure of BDP5290 with MRCKβ also shows the conservation of the inner water molecule at the gatekeeper, forming a water bridge between Thr137 and the carbonyl of the amide on BDP5290, which acts as a hydrogen bond acceptor while the piperazine faces towards the solvent (Figure 8B). The heterocyclic ligand of the second-generation MRCK inhibitors satisfies the hydrogen bonding potential of water inside the binding pocket. The pyrimidine of BDP9066 (Figure 8A) has an advantage over the thiazole ligand of BDP8900 (Figure 8C) and the chloropyrazole of BDP5290 (Figure 8B), as the second nitrogen in the ring forms a water bridge to hydrogen bond with Lys105 [31,52]. The most intriguing component of modern inhibitors is the diazaspirocycle, which is the solvent-exposed part of the molecule. The diazaspirocycle can form hydrogen bonds with water molecules due to the extensive reach and geometry of the nitrogen on the second piperidine ring. 

## 7. MRCK Inhibitors in Biological Studies

Despite the unclear mechanism of action as an MRCK inhibitor [47] and numerous additional effects on cells, including reactive oxygen species generation [54], DNA intercalation [55], and the inhibition of acetylcholinesterases [56] and BCL-XL [57], chelerythrine has been used as a tool compound to examine the biological functions of MRCK. The treatment of HeLa cells with chelerythrine chloride led to changes in F-actin organization and inhibited cell migration, in agreement with an effect on inhibiting MRCK activity [47]. In agreement with these observations, the effect of the treatment of HeLa cells with the diacylglycerol analogue bis(3-trifluoromethylbenzyl) 5-(hydroxymethyl)isophthalate (HMI-1a3) on cell morphology, proliferation, and actin structures was reversed by chelerythrine chloride and not protein kinase C inhibitors or protein knockdown, consistent with its role in MRCK signalling. Chelerythrine was recently reported to negatively regulate the cell-surface localization of the adenosine triphosphate-binding cassette transporter 4 (ABCB4), likely through MLC phosphorylation [58].

The treatment of a range of cancer cell lines (including H522 and A549 lung, PANC-1 pancreas, MDA-MB-231 breast, and A375M melanoma) with DJ4 reduced MLC and MYPT1 phosphorylation, altered cytoskeleton organization, and inhibited cell migration and invasion [52]. Given that DJ4 is a potent inhibitor of ROCK kinases, as well as MRCK kinases, it is not clear what the respective contribution of each type of kinase to these effects was.

The more potent and selective inhibitor BDP5290 was similarly shown to reduce MLC phosphorylation, alter cytoskeleton organization, and block the migration of and invasion by MDA MB 231 human breast cancer cells [53]. The polarization of epithelial cells in MDCK monolayers was found to be sensitive to BDP5290 [59], supporting a role for MRCK in this process. In addition, BDP5290 was recently used to validate a role for MRCK in phagocytosis in retinal pigment epithelial cells [60].

The most potent and selective BDP9066 inhibitor has been used to demonstrate a role for MRCK in cancer. MDA MB 231 breast cancer cells and SCC12 squamous cell carcinoma treated with BDP9066 displayed morphological changes and altered cytoskeleton organization, and were reduced in the migration and invasion activities [32]. Consistent with the observed elevation in MRCK expression in cutaneous squamous cell carcinoma (SCC) relative to normal human skin, the topical application of BDP9066 reduced tumour growth, but not tumour initiation, in a chemically induced mouse model of skin cancer [32]. Furthermore, the immunohistochemical staining of tumour sections with an antibody that recognized the autophosphorylated Ser1003 of MRCKα as a biomarker of activity status revealed that BDP9066 was effective at reducing kinase activity.

Glioma cells subjected to clinically relevant levels of ionizing radiation had increased levels of phosphorylated MLC and migrated more rapidly, both of which were blocked by BDP9066 [42]. Importantly, the radiation-induced spread of orthotopic glioma brain tumours was blocked by systemic BDP9066 treatment, while the combination of radiotherapy plus systemic BDP9066 had a greater effect on the survival of tumour-bearing mice than either treatment alone [42]. Taken together, these findings indicate that MRCK inhibition could have clinical value in combination with radiotherapy for glioma patients.

Given the evidence of increased MRCK expression in ovarian cancers, the sensitivity of HGSOC cell lines to MRCK inhibition was examined. Treatment with 1 µM of BDP9066 reduced the viability of 7/9 established HGSOC cell lines by more than 50%, a concentration that also significantly reduced colony formation (8/9 cell lines) and spheroid growth in 3D (5/7 cell lines) [44]. In support of these observations, MRCKα knockdown by siRNA resulted in 6/10 HGSOC cell lines having >50% loss of viability, with evidence of caspase activation in 7/10 cell lines [44]. Taken together, these results demonstrate the sensitivity of established HGSOC cell lines to MRCK inhibition and demonstrate the potential for MRCK inhibitors as chemotherapeutic agents that work by targeting cell proliferation and survival, in addition to blocking migration and invasion.

## 8. Conclusions

Despite their near-contemporaneous discovery, our knowledge of the biological functions of the ROCK kinases is vastly greater than that of the MRCK kinases. A major reason for this difference is due to the early discovery of pharmacological ROCK inhibitors [13] that have been important chemical biology tools for over 25 years. Although there have been concerns about their selectivity, for example, the ROCK inhibitor Y-27632 was reported to be comparably effective towards PRK2 [61], the availability of multiple inhibitors with divergent pharmacophores has enabled thorough pharmacological investigation of ROCK functions, which can also be validated by knockdown or knockout approaches. The first reported MRCK inhibitor was chelerythrine chloride [47], which was originally characterized as a potent protein kinase C inhibitor [49]. Given the broad specificity and unknown mechanism of action, chelerythrine chloride has not been widely utilized. The 2-pyridyl pyrazole amide inhibitor BDP5290 [53] and the 7-azaindole inhibitors BDP8900 and BDP9066 [32] are improvements over chelerythrine chloride, given their clear mechanism of action, high potency, and selectivity, with BDP9066 being particularly useful as a chemical biology tool. Our knowledge of MRCK’s functions in normal cells and roles in diseases including cancer will undoubtedly be greatly increased, and the relative importance of MRCK versus ROCK as the mediators of phosphorylation of key substrates, such as the regulatory myosin light chains, will be determined.

## Figures and Tables

**Figure 1 cells-12-00534-f001:**
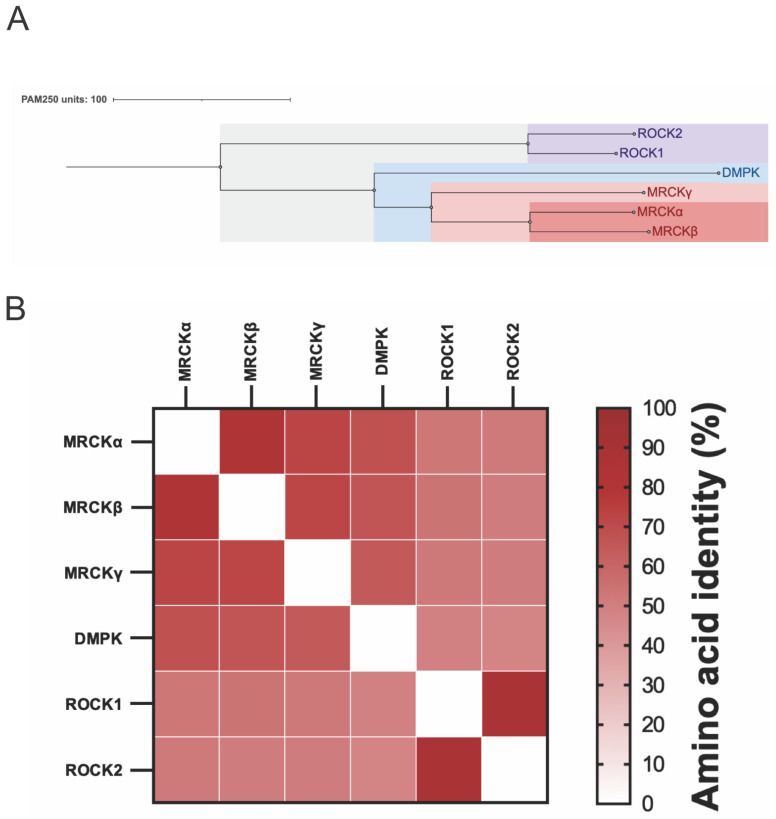
**Relatedness of MRCK, ROCK, and DMPK kinases**. (**A**). The ROCK1, ROCK2, MRCKα, MRCKβ, MRCKγ, and DMPK kinase domains were aligned using protein sequence data from UniProt [16] and ClustalOmega [17]. The results were presented using Jalview [18], and the PAM250 matrix was used to convert the similarity percentages into average distances measured to represent evolutionary distances among the proteins [19]. (**B**). Pairwise alignment of the ROCK1, ROCK2, MRCKα, MRCKβ, MRCKγ, and DMPK kinase domains produced percentage identity scores that highlighted the similarities between each kinase domain. These results were graphed as a heat map of their pairwise amino acid identities.

**Figure 2 cells-12-00534-f002:**
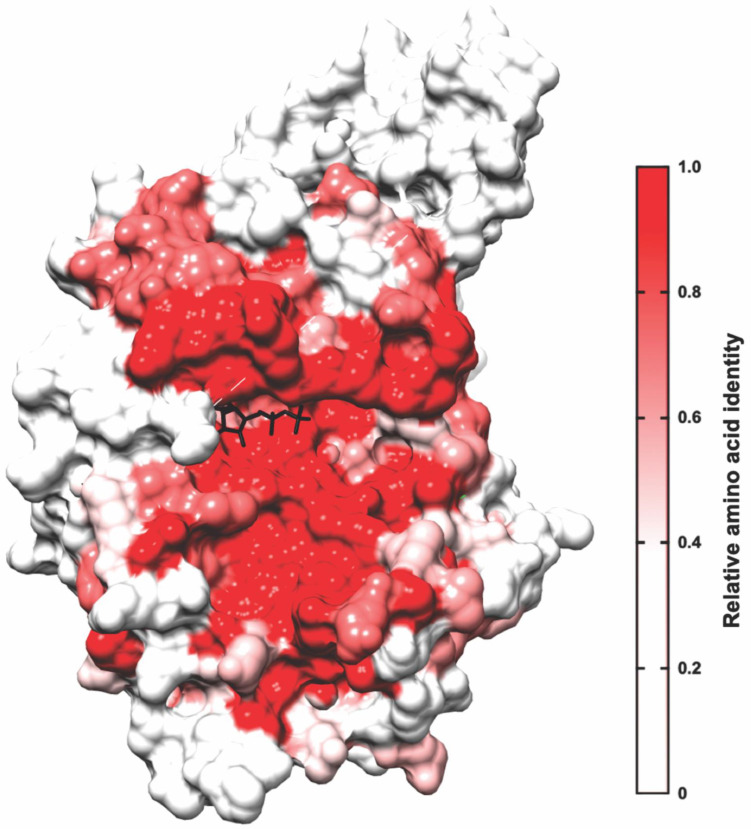
**Percentage amino acid identities of kinase domains mapped onto MRCKβ**. ROCK1, ROCK2, MRCKα, MRCKβ, MRCKγ, and DMPK kinase domains were aligned, and the calculated amino acid identity scores between all six kinases were used to create a heat map on the 3D structure of MRCKβ bound to ADP (black; PDB 4UAK) [20] with UCSF Chimera [21]. The minimum relative amino acid identity threshold was set to 0.5, where any residues below the threshold were coloured white and residues with higher percentage identity scores were coloured in varying shades of red, as indicatedphosphorylate and activate LIMK2, although the role of LRAP25 was not examined [22]. Together, these phosphorylation events catalysed by MRCK kinases lead to the stabilization of actin filaments and increased MLC phosphorylation, and the consequent generation of contractile force, which collectively promote cell motility (Figure 3).

**Figure 3 cells-12-00534-f003:**
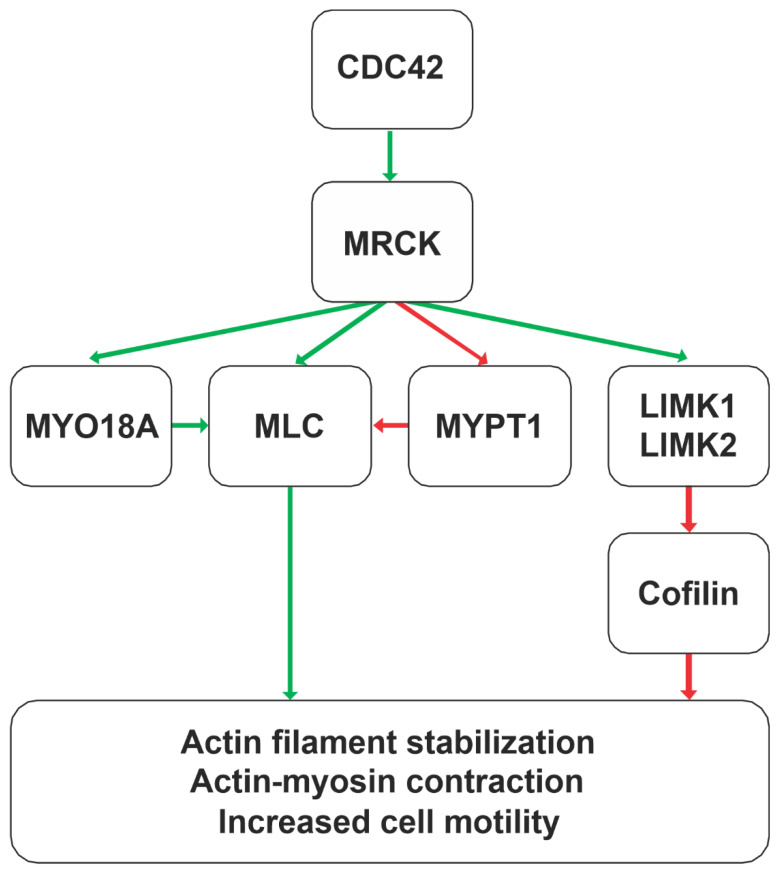
**MRCK signalling pathways leading to actin–myosin cytoskeleton regulation**. Acting downstream of the Cdc42 GTPase, the MRCK kinases MRCKα, MRCKβ, and MRCKγ phosphorylate substrates that ultimately lead to the stabilization of actin filaments and promotion of actin–myosin contractile force generation. These actions on the actin–myosin cytoskeleton contribute to increased cell motility.

**Figure 4 cells-12-00534-f004:**
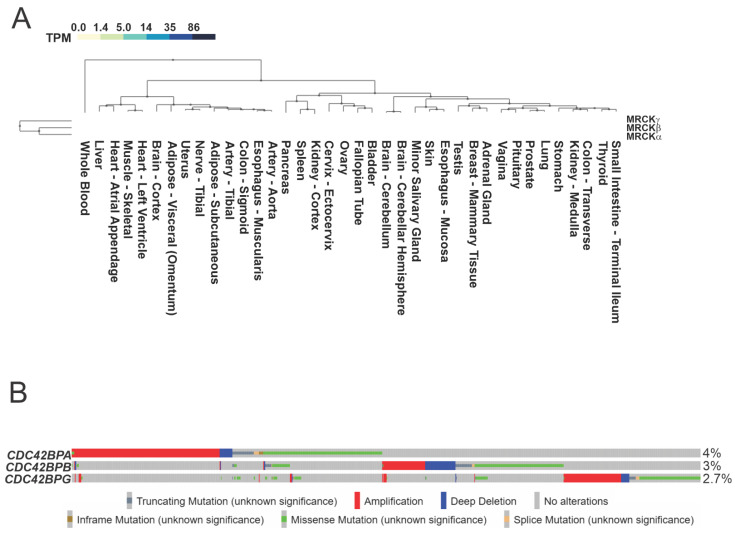
**MRCK expression and gene mutations**. (**A**). The Genotype–Tissue Expression (GTEx) RNA-Seq database collected from non-diseased tissues across nearly 1000 individuals was queried for the expression levels of MRCKα, MRCKβ, and MRCKγ. Values indicate the number of transcripts per million (TPM) for each mRNA. (**B**). The Cancer Genome Atlas program was queried via cBioPortal for genomic alterations to the genes encoding Cdc42BPA, Cdc42BPB, and Cdc42BPG in the TCGA PanCancer Atlas Studies of 32 cancer types.

**Figure 5 cells-12-00534-f005:**
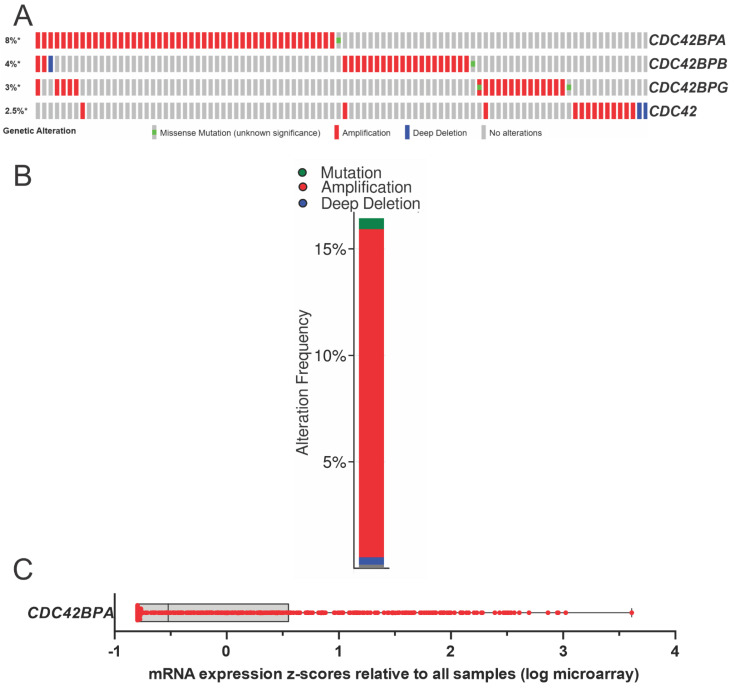
**MRCK expression and gene mutations in ovarian cancers**. (**A**). The Cancer Genome Atlas program was queried via cBioPortal for genomic alterations to the genes encoding Cdc42BPA, Cdc42BPB, and Cdc42BPG, and their upstream regulator Cdc42 in the TCGA studies of ovarian cancer [45,46]. (**B**). Cumulative plot of the genomic alterations to Cdc42BPA, Cdc42BPB, Cdc42BPG, and Cdc42. * indicates that values have been rounded to the nearest integer value. (**C**). Elevated MRCKα mRNA expression in ovarian cancer tumours determined by microarray. Log-transformed mRNA expression z-scores compared with the expression distribution of all samples, as determined with Agilent microarrays for 558 serous ovarian cancer patient samples in the TCGA Firehose database. The box indicates the upper and lower quartiles, with the line indicating median values.

**Figure 6 cells-12-00534-f006:**
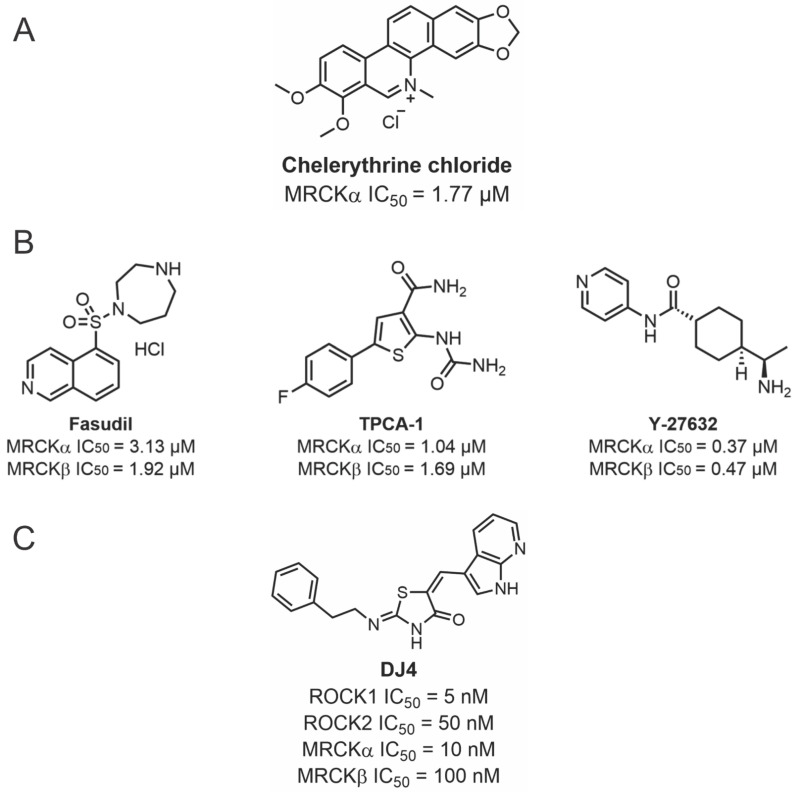
**Structures of chelerythrine chloride, Fasudil, TPCA-1, Y-27632, and DJ4 and their inhibitory activities**. (**A**). Structure of chelerythrine chloride and its inhibitory activity against MRCKα. (**B**). Structure of Fasudil, TPCA-1, and Y-27632 and their inhibitory activities against MRCKα and MRCKβ. (**C**). Structure of DJ4 and its inhibitory activity against ROCK1, ROCK2, MRCKα, and MRCKβ.

**Figure 7 cells-12-00534-f007:**
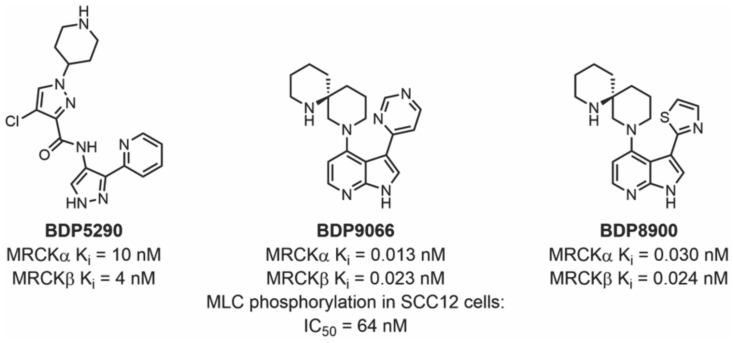
**Structures of BDP5290, BDP9066, and BDP8900 with their corresponding enzymatic and cellular activities**.

**Figure 8 cells-12-00534-f008:**
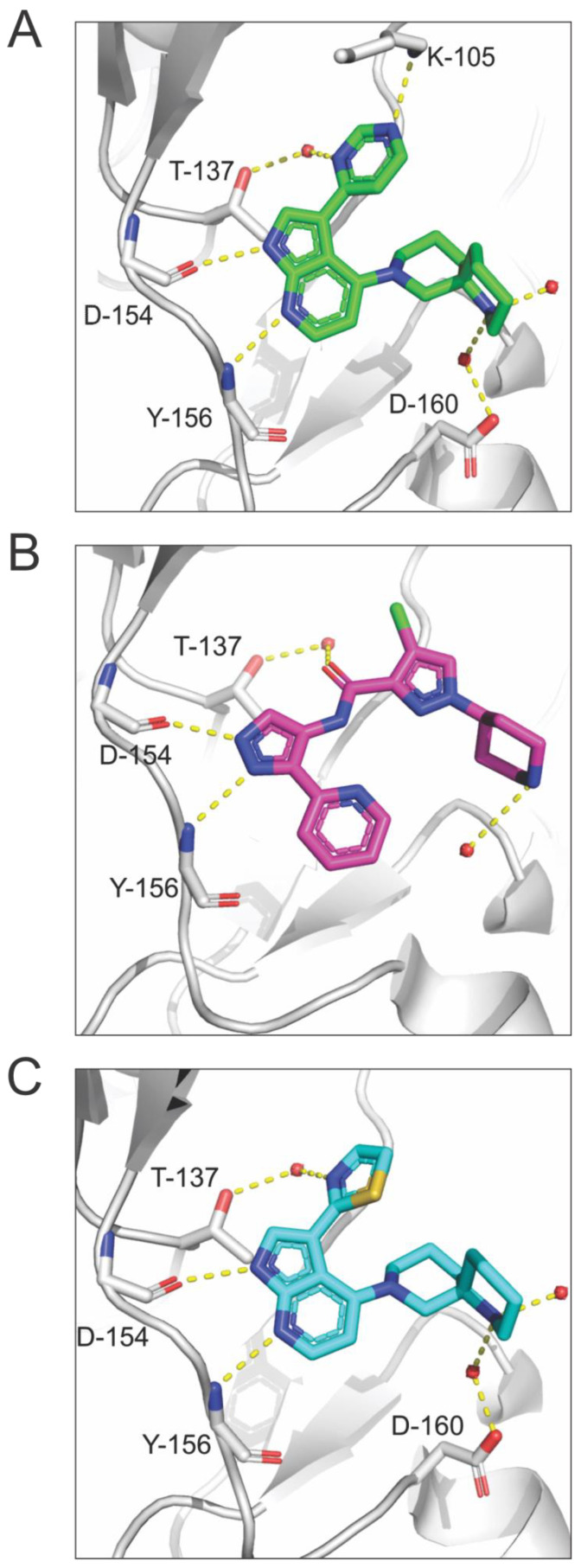
**X-ray co-crystal structures of inhibitors and MRCKβ**. (**A**). Co-crystal structure of the MRCKβ kinase domain in complex with BDP9066 (PDB 5OTF) [20]. (**B**). Co-crystal structure of the MRCKβ kinase domain in complex with BDP5290 (PDB 4UAL) [20]. (**C**). Co-crystal structure of the MRCKβ kinase domain in complex with BDP8900 (PDB 5OTE) [28]. Protein residues are represented by a single-letter amino acid code and residue number. Water molecules are represented by red spheres and hydrogen bonds are highlighted with yellow dotted lines.

## Data Availability

Not applicable.

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
