# Peer review of "Opportunities and Challenges for the Development of MRCK Kinases Inhibitors as Potential Cancer Chemotherapeutics"

_cells, 2023, doi:10.3390/cells12040534_

Round 1

Reviewer 1 Report

 In the present manuscript, the author depicted the application of MRCK Kinases Inhibitors as potential cancer chemotherapeutics.

Comment:

Please provide more clinical application cases to support the conclusion.

Author Response

Reviewer 1

Please provide more clinical application cases to support the conclusion.

RESPONSE: There are no MRCK inhibitors in the clinic at this point in time, so there are no additional cases that can be added to the manuscript.

Reviewer 2 Report

In the present review, discuss the potential benefits of the pharmacological inhibition of MRCKs in cancer therapy. After an overview on MRCK functions in generating the contractile force that in turn promotes cell motility in normal epithelial cells, the authors describe the associations of genomic alterations of  MRCK genes with cancer growth and progression. The authors also summarize the structure and activity of current MRCKs inhibitors by comparing their selectivity and specificity to ROCK. Finally, they describe the two recent potent and selective inhibitors of MRCKs, BDP8900 and BDP9066, by evidencing their biological activity as chemotherapeutic agents.

The review is well written and exhaustive.

Minor points:

Line 212: in the discussion on the data showing that chelerythrine chloride inhibited PKC, a further reference should be added (Lin et al. Sci Rep 2017 7; 2022. Doi: 10.1038/s41598-017-02222-0).

Line 222: TPCA-1 has been also described to inhibit STAT3 and IKK/NF-kB. This work should be cited (Nan et al.  Mol Cancer Ther. 2014 Mar;13(3):617-29.doi: 10.1158/1535-7163.MCT-13-0464)

Author Response

Reviewer 2

Line 212: in the discussion on the data showing that chelerythrine chloride inhibited PKC, a further reference should be added (Lin et al. Sci Rep 2017 7; 2022. Doi: 10.1038/s41598-017-02222-0).

RESPONSE: Reference was added.

Line 222: TPCA-1 has been also described to inhibit STAT3 and IKK/NF-kB. This work should be cited (Nan et al.  Mol Cancer Ther. 2014 Mar;13(3):617-29.doi: 10.1158/1535-7163.MCT-13-0464)

RESPONSE: Reference was added.

Reviewer 3 Report

The review presented by Ruscetta et al. is a well organised and complete review of the structure and function of the myotonic dystrophy-related Cdc42-binding kinases (MRCK). Also, they analysed the expression of MRCKs in different tissues and their relationship with cancer development. The authors described the structure and mechanism of action of three inhibitors for MRCK kinases and one inhibitor with dual activity for MECK and ROCK kinases. Also, the structure and function of dual ROCK and MRCK kinase inhibitors to target both pathways simultaneously, such as BDP5290, BDP9066 and BDP8900. These inhibitors could serve as the basis for developing more potent and specific inhibitors for these kinases and could be useful in the future for cancer treatment. This paper adds interesting and helpful information to the existing literature. The authors logically describe the information and discuss the ideas on the role of the MRCKs on actin-myosin cytoskeleton organization and dynamics in normal cells and their roles in cancer cell viability, migration and invasion and the importance of having selective inhibitors for these kinases.

The references are appropriate and up-to-date. The references are related to the main topic of the article.

Minor points to be addressed:

1. The paragraph, lines 271-272, is not clear. Please explain.

2. In figure 8 panel A the letter D for aspartic 154 is missing in the image. In panel B, the aspartic 160 is missing; or it is unnecessary for the interaction with the inhibitor. Please explain further in the text.

3. References 55 to 59 are not indicated in the text.

Author Response

Reviewer 3

  1. The paragraph, lines 271-272, is not clear. Please explain.

RESPONSE: The first three sentences in the “Binding Mode of MRCK Inhibitors” section have been re-written to improve their clarity.

  1. In figure 8 panel A the letter D for aspartic 154 is missing in the image. In panel B, the aspartic 160 is missing; or it is unnecessary for the interaction with the inhibitor. Please explain further in the text.

RESPONSE: The missing letter D for Asp 154 in Figure 8A has been added. There is no interaction with aspartic acid 160 by the compound in Figure 8B, which is why it wasn’t indicated. This lack of interaction has been noted in the text.

  1. References 55 to 59 are not indicated in the text.

RESPONSE: The references were originally found in lines 96 to 107 but have shifted with the edits. See figure 1 and 2. They are also not 55-59, now 20-25 with correct organization.